# Descriptive epidemiology of hospitalized patients with bacterial nosocomial pneumonia who experience 30-day readmission in the US, 2014–2019

Marya D. Zilberberg[1]*, Brian H. Nathanson[2], Laura A. Puzniak[3], Noah W. D. Zilberberg[1,4], Andrew F. Shorr[5]

1 EviMed Research Group, LLC, Goshen, MA, United States of America, 2 OptiStatim, LLC, Longmeadow, MA, United States of America, 3 Merck & Co., Inc., Kenilworth, NJ, United States of America, 4 Universty of Massachusetts, Amherst, MA, United States of America, 5 Washington Hospital Center, Washington, DC, United States of America

* evimedgroup@gmail.com

## Abstract

### Introduction

Nosocomial pneumonia (NP) remains associated with excess morbidity and mortality. The effect of NP on measures such as re-admission at 30 days remains unclear. Moreover, differing types of NP may have varying impacts on re-admissions.

### Methods

We conducted a multicenter retrospective cohort study within the Premier Research database, a source containing administrative, pharmacy, and microbiology data. We compared NP patients readmitted with pneumonia (RaP) as the principal diagnosis to those readmitted for other reasons (RaO) with respect to the type of NP (ventilator-associated bacterial pneumonia [VABP], ventilated hospital-acquired bacterial pneumonia [vHABP], and non-ventilated HABP [nvHABP]), and characteristics and outcomes of the index hospitalization.

### Results

Among 17,819 patients with NP, 14,123 (79.3%) survived to discharge, of whom 2,151 (15.2%) required an acute readmission within 30 days of index discharge. Of these, 106 (4.9%) were RaP, and the remainder were RaO. At index hospitalization, RaP patients were older (mean age [SD] 67.4 [13.9] vs. 63.0 [15.2] years), more likely medical (44.3% vs. 36.7%), and less chronically ill (median [IQR] Charlson scores (3 [2–5] vs. 4 [2–5]) than persons with RaO. Bacteremia (10.4% vs. 17.5%), need for vasopressors (15.1% vs. 20.0%), dialysis (9.4% vs. 16.5%), and/or sepsis (9.4% vs. 16.5%) or septic shock 14.2% vs. 17.1%) occurred less frequently in the RaP group. With respect to NP type, nvHABP was most common in RaP (47.2%) and VABP in RaO (38.1%).

**Data Availability Statement:** Premier is a proprietary database available for purchase to any investigator. Due to the legal constraints of

licensing, the authors are unable to upload the data used. However, the authors have provided a link that includes an infographic and a lwhitepaper which contains very specific information related to the data in the PHD. https://www.premierinc.com/newsroom/education/what-fuels-pinc-ai-healthcare-data The authors had no special privileges in accessing these datasets which other interested researchers would not have. Data access queries may be directed to Jim Sianis, Senior Director at Premier Inc. (james_sianis@premierinc.com).

**Funding:** Study supported by a grant from Merck Sharp & Dohme Corp., a subsidiary of Merck & Co., Inc., Kenilworth, NJ, USA. • MDZ's employer, EviMed Research Group, LLC, has received research grant support from Merck Sharp & Dohme Corp., a subsidiary of Merck & Co., Inc., Kenilworth, NJ, USA. • BHN's employer, OptiStatim, LLC, has received support from EviMed Research Group, LLC, Goshen, MA. • At the time of the study conduct, LAP was an employee of Merck Sharp & Dohme Corp., a subsidiary of Merck & Co., Inc. Kenilworth, NJ, USA and a stockholder in Merck & Co., Inc., Kenilworth, NJ, USA. • NWDZ has no conflicts to report. • AFS is a consultant to and has received research grant support from Merck Sharp & Dohme Corp., a subsidiary of Merck & Co., Inc., Kenilworth, NJ, USA. • Other than LAP's participation in the study, the funders had no role in study design, data collection and analysis, decision to publish, or preparation of the manuscript.

**Competing interests:** I have read the journal's policy. In addition to funding statement above, the following potential competing interest exist: •MDZ and AFS have received grant support and/or have served as consultants to Lungpacer, Melinta, Tetraphase, Pfizer, Astellas, Shionogi, The Medicines Company, Spero, and Theravance.

## Conclusions

One in seven survivors of a hospitalization complicated by NP requires an acute rehospitalization within 30 days. However, few of these readmissions had a principal diagnosis of pneumonia, irrespective of NP type. Of the 5% of NP subjects with RaP, the plurality initially suffered from nvHABP.

## Introduction

Nosocomial pneumonia (NP), which includes both hospital-acquired (HAP) and ventilator-associated (VAP), continues to be a frequent hospital-acquired complication (HAC). In fact, NP accounts for over one-fifth of all healthcare-associated infections (HAIs) [1]. Presently, the Centers for Medicare and Medicaid Services (CMS), via its Hospital Value Based Purchasing program, target both HACs and 30-day readmissions as part of its initiatives focusing on quality improvement [2]. As both a HAC and a syndrome associated with potential readmissions, NP represents a key component of efforts to improve healthcare quality and to reduce healthcare expenditures. Much effort has rightly been devoted to NP prevention, and to the optimization of its treatment and outcomes. For example, the ventilator bundle serves as one of the most widely used strategies for avoiding VAP [3]. Likewise, appropriate empiric treatment for NP is now well understood as the main modifiable factor affecting both mortality and morbidity [4–8]. Despite improvements in the understanding of what drives the development of NP and the outcomes associated with it, less is known regarding the variables that affect hospital readmission after a bout of NP. Specifically, it remains unclear whether a readmission following NP is driven mainly by the NP event itself or by other comorbid conditions. This distinction has important implications. Namely, if there is a considerable risk of being readmitted with pneumonia after an admission complicated by HAP/VAP it suggests that some aspect of NP care may be implicated as a cause. Conversely, if patients with NP are readmitted primarily for non-pneumonia conditions, then efforts to address readmissions should focus on issues other than the initial pneumonia care.

To explore this topic, we analyzed patients who survived a case of NP and were subsequently readmitted to the hospital, and compared those who were readmitted for pneumonia (RaP) to those whose rehospitalization was due to other reasons (RaO).

## Methods

### Ethics statement

Because this study used already existing fully de-identified data, it was exempt from ethics review under US 45 CFR 46.101(b)4 [9].

### Study design and patient population

We conducted a multi-center retrospective cohort study of hospitalized patients with culture-positive non-ventilated hospital-acquired bacterial pneumonia (nvHABP), ventilated hospital-acquired bacterial pneumonia (vHABP), or ventilator-associated pneumonia (VABP) to explore the incidence of 30-day hospital readmissions specifically for pneumonia versus other causes. A RaP was defined as a rehospitalization with pneumonia as the principal diagnosis. A RaO was one with the principal diagnosis other than pneumonia.

The case identification approach relied on a slight modification of a previously published algorithm, and the details of the current study methods have been published elsewhere [10–12].

Briefly, patients were included if they were adults (age ≥ 18 years) whose initial pneumonia appeared as a secondary diagnosis, whose index respiratory and/or blood culture had to be obtained on hospital day 3 or later for HABP, or on MV day 3 or later for VABP, and who were treated with an antibiotic on the day of the index culture and for the next ≥3 consecutive days. We excluded patients who fit the definition for either a complicated urinary tract infection or a complicated intra-abdominal infection in order to reduce misclassification [13,14].

## Data source

The data source was the Premier Research database, an electronic laboratory, pharmacy and billing data repository, for years 2014 through the 3[rd] quarter of 2019. The database has been described in detail previously [10–17]. Approximately 200 US institutions submitted microbiology data during the study time frame. The details of the current cohort can be found in citations #11 and #12 [11,12].

## Pneumonia classification

Pneumonia was defined as HABP if at the time of the index culture the patient was not on MV and VABP if at the time of the index culture the patient had been on MV for 3+ days. HABP was further subdivided into vHABP and nvHABP. Specifically, vHABP designation was given for patients who needed MV ≤5 days following the onset of index HABP episode and nvHABP if MV was not required.

## Infection and treatment variables

ICU admission, presence of severe sepsis or septic shock, dialysis, and vasopressor use were used as markers for acute disease severity. We determined whether each patient's admission was due to medical, surgical, or trauma diagnosis, as well as whether the patient had suffered an acute neurologic insult [18,19].

## Microbiology

We examined common Gram-positive and Gram-negative pathogens that cause bacterial nosocomial pneumonia. Namely, the Gram-negative organisms of interest were *Pseudomonas aeruginosa*, *Escherichia coli*, *and Klebsiella pneumoniae*, and *Acinetobacter baumannii*, while the Gram-positive ones were *Staphylococcus aureus* (both methicillin-susceptible [MSSA] and methicillin-resistant [MRSA]), and *Streptococcus pneumoniae*.

Antimicrobial coverage was considered appropriate if a drug administered within two days of the index culture being obtained covered the recovered organism. All other treatment was defined as inappropriate empiric treatment (IET).

## Statistical analyses

We report descriptive statistics to compare patients with RaP to those with RaO within 30 days of the index discharge across all demographics, comorbidities, infection characteristics, hospital characteristics and processes, and hospital outcomes. Continuous variables are reported as means with standard deviations (SD) and as medians with interquartile ranges (IQR). Differences between mean values were tested via the Student's t-test, and between medians using the Mann-Whitney test. Categorical data are summarized as counts and proportions, with the Chi-square test used to examine inter-group differences unless a cell count was < 5, wherein the Fisher exact test was used. P-values <0.05 were considered statistically significant.

## Results

Among 17,819 patients who met enrollment criteria, 14,123 (79.3%) survived to discharge. Of those, 2,151 (15.2%) required a rehospitalization within 30 days of discharge. Among these readmissions, 106 (4.9% of all readmissions and 0.8% of all survivors) were classified as RaP and 2,045 (95.1%) as RaO. Of those with RaO, under 10% had a principal diagnosis of either sepsis or respiratory failure, events that may have been incited by a respiratory infection (Table 1 in S1 Appendix).

All results that follow pertain to the index hospitalization involving the NP bout in these two groups.

There were few substantive differences in hospital characteristics between the RaP and RaO groups, and none reached statistical significance (Table 1). However, hospitals that treated RaP patients tended to be smaller and less likely academic than those treating RaO. While RaP patients were older than those with RaO (mean age 67.4+/-13.9 vs. 63.0+/-15.2, respectively, p<0.001), all other demographic characteristics did not differ between these groups (Table 1). Although there were some imbalances in the distribution of individual chronic comorbid conditions, the mean Charlson scores were similar in the two groups (3.92+/-2.84 in RaP vs. 3.88 +/-2.72 in RaO, p = 0.898) (Table 2 in S1 Appendix, Table 1).

Characteristics of the index hospitalization involving NP are listed in Table 2. Though not statistically significant, nvHABP was far more and VABP far less prevalent among the group with RaP, while the frequency of vHABP was nearly identical between the two groups. Although the frequencies of some individual pathogens were low, there were several that tended to differ between the groups. Most notably, *E. coli* was >2x as prevalent in RaO (11.7%) as in RaP (5.7%, p = 0.059). Conversely, *P. aeruginosa* was directionally more common in the RaP group (24.5%) than the RaO (20.1%, p = 0.267). MRSA, on the other hand, was quite common in both, and >50% relatively more likely in the RaP (22.6%) than RaO (14.9%, p = 0.037) group. The situation was similar for antimicrobial treatment prior to the index admissions, where 23.6% of the RaP and 17.0% of the RaO group had undergone antibiotic treatment within 90 days prior to that hospitalization (p = 0.079). Finally, though numerically patients with RaP were less likely to have the pathogen isolated from blood and more likely from sputum, the differences failed to reach statistical significance The acute illness severity measures during index hospitalization, on the other hand, were uniformly higher in the RaO group (Fig 1). However, all but ICU admission and septic shock failed to reach statistical significance. Specifically, there was no difference between the group in their likelihood of receiving IET.

During the index admission, clinical complications did not differ substantively between those readmitted for pneumonia as opposed to some other reason (Fig 2). Namely, the rates of incident CDI (4.7% RaP vs. 3.2% RaO, p = 0.403), extubation failure (11.3% RaP vs. 10.9% RaO, p = 0.906), and reintubation (4.7% RaP vs. 5.6% RaO, p = 0.692), were similar. Some utilization outcomes, on the other hand, were worse in the RaO than the RaP group. While the ICU LOS, did not differ between the two groups, median hospital LOS and both mean and median hospital costs were higher in the RaO than RaP (Fig 3A and 3B). In contrast, MV duration was longer in the RaP than in the RaO group during the index hospitalization (Fig 3A & 3B).

## Discussion

We demonstrate that among patients with nosocomial pneumonia who survive to be discharged, 15% require a readmission within 30 days of the index discharge. Of those 15%, only approximately one in 20 is rehospitalized specifically *for* the treatment of pneumonia. While a small percentage (<10%) of those readmitted were for other causes that may be secondary to

**Table 1. Baseline characteristics at index hospitalization among patient readmitted within 30 days of discharge.**

| | Pneumonia | | | Non-pneumonia | | | |
|---|---|---|---|---|---|---|---|
| | N | % row | % column | N | % row | % column | P value |
| N | | 106 | | | 2045 | | |
| Mean age, years (SD) | | 67.41(13.92) | | | 62.96(15.18) | | <0.001 |
| Gender: male | 66 | 5.12% | 62.26% | 1223 | 94.88% | 59.80% | 0.614 |
| Race | | | | | | | |
| White | 81 | 5.13% | 76.42% | 1499 | 94.87% | 73.30% | 0.248 |
| Black | 12 | 3.26% | 11.32% | 356 | 96.74% | 17.41% | |
| Hispanic | 2 | 3.17% | 1.89% | 61 | 96.83% | 2.98% | |
| Other | 12 | 6.38% | 11.32% | 176 | 93.62% | 8.61% | |
| Unknown | 1 | 6.67% | 0.94% | 14 | 93.33% | 0.68% | |
| Admission Source | | | | | | | |
| Non-healthcare facility (including from home) | 89 | 4.99% | 83.96% | 1694 | 95.01% | 82.84% | 0.665 |
| Clinic | 12 | 6.22% | 11.32% | 181 | 93.78% | 8.85% | |
| Transfer from ECF | 2 | 4.00% | 1.89% | 48 | 96.00% | 2.35% | |
| Transfer from another non-acute care facility | 3 | 4.17% | 2.83% | 69 | 95.83% | 3.37% | |
| Other | 0 | 0.00% | 0.00% | 53 | 100.00% | 2.59% | |
| Admission type | | | | | | | |
| Medical | 47 | 5.89% | 44.34% | 751 | 94.11% | 36.72% | 0.126 |
| Surgical | 59 | 4.40% | 55.66% | 1281 | 95.60% | 62.64% | |
| Neurologic | 9 | 4.84% | 8.49% | 177 | 95.16% | 8.66% | 0.953 |
| Trauma | 21 | 4.71% | 19.81% | 425 | 95.29% | 20.78% | 0.810 |
| Charlson Comoribidity Score | | | | | | | |
| 0 | 6 | 3.57% | 5.66% | 162 | 96.43% | 7.92% | 0.812 |
| 1 | 17 | 6.59% | 16.04% | 241 | 93.41% | 11.78% | |
| 2 | 15 | 4.73% | 14.15% | 302 | 95.27% | 14.77% | |
| 3 | 16 | 4.86% | 15.09% | 313 | 95.14% | 15.31% | |
| 4 | 15 | 4.98% | 14.15% | 286 | 95.02% | 13.99% | |
| 5+ | 37 | 4.76% | 34.91% | 741 | 95.24% | 36.23% | |
| Mean (SD) | | 3.92(2.84) | | | 3.88(2.72) | | 0.898 |
| Median [IQR] | | 3[2–5] | | | 4[2–5] | | 0.902 |
| Hospital Characteristics | | | | | | | |
| Census region | | | | | | | |
| Midwest | 36 | 5.78% | 33.96% | 587 | 94.22% | 28.70% | 0.701 |
| Northeast | 20 | 4.38% | 18.87% | 437 | 95.62% | 21.37% | |
| South | 43 | 4.69% | 40.57% | 874 | 95.31% | 42.74% | |
| West | 7 | 4.55% | 6.60% | 147 | 95.45% | 7.19% | |
| Number of Beds | | | | | | | |
| <100 | 2 | 8.70% | 1.89% | 21 | 91.30% | 1.03% | 0.059 |
| 100 to 199 | 11 | 6.79% | 10.38% | 151 | 93.21% | 7.38% | |
| 200 to 299 | 18 | 6.84% | 16.98% | 245 | 93.16% | 11.98% | |
| 300 to 399 | 14 | 7.00% | 13.21% | 186 | 93.00% | 9.10% | |
| 400 to 499 | 23 | 5.45% | 21.70% | 399 | 94.55% | 19.51% | |
| 500+ | 38 | 3.52% | 35.85% | 1043 | 96.48% | 51.00% | |
| Teaching | 62 | 4.41% | 58.49% | 1345 | 95.59% | 65.77% | 0.124 |
| Urban | 93 | 4.77% | 87.74% | 1856 | 95.23% | 90.76% | 0.298 |
| Index pneumonia type | | | | | | | |

*(Continued)*

**Table 1.** (Continued)

| | Pneumonia | | | Non-pneumonia | | | |
|---|---|---|---|---|---|---|---|
| | N | % row | % column | N | % row | % column | P value |
| vHABP | 24 | 4.90% | 22.64% | 466 | 95.10% | 22.79% | 0.123 |
| nvHABP | 50 | 6.02% | 47.17% | 780 | 93.98% | 38.14% | |
| VABP | 32 | 3.85% | 30.19% | 799 | 96.15% | 39.07% | |

nvHABP = non-ventilated HABP; vHABP = ventilated HABP; VABP = ventilator-associated bacterial pneumonia; SD = standard deviation; ECF = extended care facility; AIDS = acquired immune deficiency syndrome; IQR = interquartile range.

pneumonia, such as sepsis or respiratory failure, most of the 30-day readmissions are for other conditions. Interestingly, the type of NP did not seem correlated with the reason for readmission. Additionally, there were small differences in the index pathogen distributions. Specifically, MRSA proved to be significantly more common in the RaP than RaO group. Measures of illness severity were lower in those eventually readmitted with RaP than with RaO. Most notably, exposure to IET for the NP did not differ between the two groups.

Since fiscal year 2013, the Centers for Medicare and Medicaid Services have been charged with statutory enforcement of quality standards for of the care delivered. As a part of value-based purchasing, the Hospital Readmissions Reduction Program (HRRP) is designed to punish hospitals with excess 30-day readmissions by reducing their reimbursements in certain conditions [2]. As a result, institutions have focused on improving their discharge planning so as to limit patients' exposure to this undesirable outcome. However, since it is virtually impossible to eliminate these readmissions altogether, it is important to understand their prevalence and what modifiable factors influence them. While the epidemiology of 30-day readmissions has been described in community-acquired and healthcare-associated pneumonia, it has been less addressed in the setting of NP [20, 21]. In our study we found that this phenomenon occurs in 15% of all survivors of NP, and that the differences between those who get admitted for treatment of a new or a recurrent or an undertreated pneumonia are small but present during the index hospitalization. One striking implication of our results is that since RaP patients are less likely to have had an ICU admission or require vasopressors, and were no more apt to receive IET than RaO, there is little reason to conclude that the processes of care these patients received in the hospital resulted in this additional pneumonia admission. That is, undertreatment or erroneous treatment of their index NP cannot be blamed for their rehospitalization for a pneumonia. Efforts to address these readmissions, therefore, may need to focus elsewhere and on other points of intervention.

The current analysis sheds some light on who among those needing a repeat hospitalization may be at a higher risk for a pneumonia readmission. For example, there was a significantly higher prevalence of nvHABP among those with RaP than those with RaO, and, conversely, a lower prevalence of VABP as their index NP. Additionally, those with RaP had shorter and less costly index hospitalizations. While this difference could be the consequence of nvHABP's generally lower severity than vHABP's and VABP's, it could also imply a truncated acute care stay where a longer one might have benefitted the patients vis-à-vis reducing their risk for a repeat pneumonia admission. Unfortunately, the structure of the database does not allow us to disentangle these causes and effects. On the other hand, among those whose index NP required MV, the MV duration was longer in the setting of RaP than in the group with RaO. As for the NP pathogen differences, only *E. coli* (~1/2 the prevalence in RaP of the RaO) and MRSA (~1/3 more likely in RaP than RaO) appeared to have any substantive differences. It is, however,

**Table 2. Index infection characteristics, treatment, and hospital events.**

| | Pneumonia | | | Non-pneumonia | | | |
|---|---|---|---|---|---|---|---|
| | N | % row | % column | N | % row | % column | P value |
| N | | 106 | | | 2045 | | |
| Organism | | | | | | | |
| Gram negative | | | | | | | |
| Enterobacteriaceae | | | | | | | |
| *Klebsiella pneumoniae* | 9 | 3.78% | 8.49% | 229 | 96.22% | 11.20% | 0.524 |
| *Proteus mirabilis* | 2 | 3.77% | 1.89% | 51 | 96.23% | 2.49% | 1.000 |
| *Escherichia coli* | 6 | 2.44% | 5.66% | 240 | 97.56% | 11.74% | 0.059 |
| *Enterobacter cloacae* | 6 | 5.36% | 5.66% | 106 | 94.64% | 5.18% | 0.821 |
| *Providencia spp* | 0 | 0.00% | 0.00% | 8 | 100.00% | 0.39% | 1.000 |
| *Serratia marcescens* | 7 | 5.65% | 6.60% | 117 | 94.35% | 5.72% | 0.668 |
| *Morganella morganii* | 1 | 9.09% | 0.94% | 10 | 90.91% | 0.49% | 0.427 |
| *Enterobacter aerogenes* | 3 | 5.08% | 2.83% | 56 | 94.92% | 2.74% | 0.766 |
| *Proteus other* | 0 | 0.00% | 0.00% | 4 | 100.00% | 0.20% | 1.000 |
| *Citrobacter freundii* | 1 | 6.67% | 0.94% | 14 | 93.33% | 0.68% | 0.533 |
| *Klebsiella oxytoca* | 1 | 2.13% | 0.94% | 46 | 97.87% | 2.25% | 0.727 |
| *Enterobacter other* | 0 | 0.00% | 0.00% | 11 | 100.00% | 0.54% | 1.000 |
| *Citrobacter other* | 1 | 5.26% | 0.94% | 18 | 94.74% | 0.88% | 0.619 |
| *Serratia other* | 0 | 0.00% | 0.00% | 4 | 100.00% | 0.20% | 1.000 |
| *Klebsiella other* | 0 | 0.00% | 0.00% | 2 | 100.00% | 0.10% | 1.000 |
| Other gram negative | | | | | | | |
| *Pseudomonas aeruginosa* | 26 | 5.94% | 24.53% | 412 | 94.06% | 20.15% | 0.267 |
| *Acinetobacter baumannii* | 3 | 5.66% | 2.83% | 50 | 94.34% | 2.44% | 0.744 |
| *Stenotrophomonas maltophilia* | 1 | 5.26% | 0.94% | 18 | 94.74% | 0.88% | 0.619 |
| *Moraxella catarrhalis* | 1 | 50.00% | 0.94% | 1 | 50.00% | 0.05% | 0.096 |
| *Haemophilus influenzae* | 4 | 7.84% | 3.77% | 47 | 92.16% | 2.30% | 0.315 |
| Gram positive | | | | | | | |
| *Staphylococcus aureus* | 47 | 5.65% | 44.34% | 785 | 94.35% | 38.39% | 0.221 |
| MRSA | 24 | 7.32% | 22.64% | 304 | 92.68% | 14.87% | 0.037 |
| MSSA | 23 | 4.56% | 21.70% | 481 | 95.44% | 23.52% | 0.725 |
| *Streptococcus pneumoniae* | 4 | 5.13% | 3.77% | 74 | 94.87% | 3.62% | 0.792 |
| Streptococcus other | 0 | 0.00% | 0.00% | 37 | 100.00% | 1.81% | 0.258 |
| Polymicrobial | | | | | | | |
| 1 | 88 | 4.78% | 83.02% | 1752 | 95.22% | 85.67% | |
| 2 | 18 | 6.52% | 16.98% | 258 | 93.48% | 12.62% | 0.225 |
| 3+ | 0 | 0.00% | 0.00% | 35 | 100.00% | 1.71% | |
| Gram stain | | | | | | | |
| Gram negative only | 55 | 4.52% | 51.89% | 1163 | 95.48% | 56.87% | 0.448 |
| Gram positive only | 41 | 5.23% | 38.68% | 743 | 94.77% | 36.33% | |
| Both Gram positive and Gram negative | 10 | 6.71% | 9.43% | 139 | 93.29% | 6.80% | |
| Culture source | | | | | | | |
| Blood | 9 | 2.81% | 8.49% | 311 | 97.19% | 15.21% | 0.059 |
| Sputum | 64 | 5.52% | 60.38% | 1096 | 94.48% | 53.59% | 0.281 |
| Respiratory | 34 | 4.72% | 32.08% | 687 | 95.28% | 33.59% | 0.820 |
| Antibiotics within 90 days prior to admission | 25 | | 23.58% | 347 | | 16.97% | 0.079 |
| Empiric treatment appropriateness | | | | | | | |
| Non-IET | 87 | 4.93% | 82.08% | 1677 | 95.07% | 82.00% | 0.605 |
| IET | 7 | 3.76% | 6.60% | 179 | 96.24% | 8.75% | |
| Indeterminate | 12 | 5.97% | 11.32% | 187 | 93.03% | 9.14% | |

MRSA = methicillin-resistant *S. aureus*; MSSA = methicillin-susceptible *S. aureus*; SD = standard deviation; IQR = interquartile range; ICU = intensive care unit;

MV = mechanical ventilation; IET = inappropriate empiric therapy.

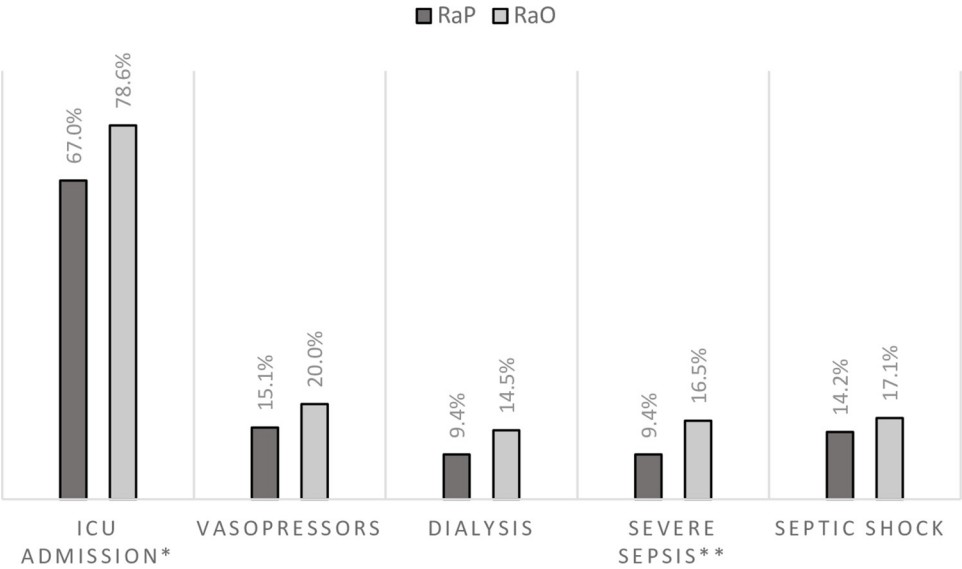

**Fig 1. Index hospitalization illness severity measures.** RaP = readmission for pneumonia; RaO = readmission for another condition. *P = 0.005. **P = 0.053.

notable too that receiving antibiotics within 90 days of the index hospitalization was also far more common in the RaP than the RaO group.

What should be made of our findings? One possible use is to help identify NP patients who may be at an increased risk for a RaP, such as those with nvHABP or those with MRSA. The importance of MRSA is unclear in that most of these patients received initial appropriate antibiotic therapy. It may be that MRSA serves as a surrogate marker for other unmeasured patient characteristics that may be associated with risk for readmission. Though nothing about our data suggests what interventions could prevent such readmissions, it is possible that more

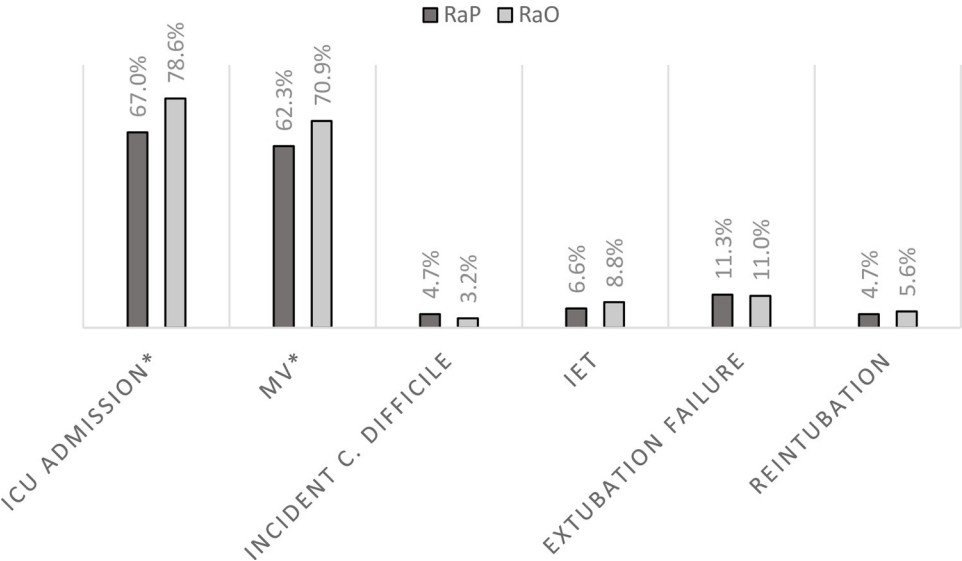

**Fig 2. Index hospitalization complications.** RaP = readmission for pneumonia; RaO = readmission for another condition; ICU = intensive care unit; MV = mechanical ventilation. *P = 0.005. **P = 0.059.

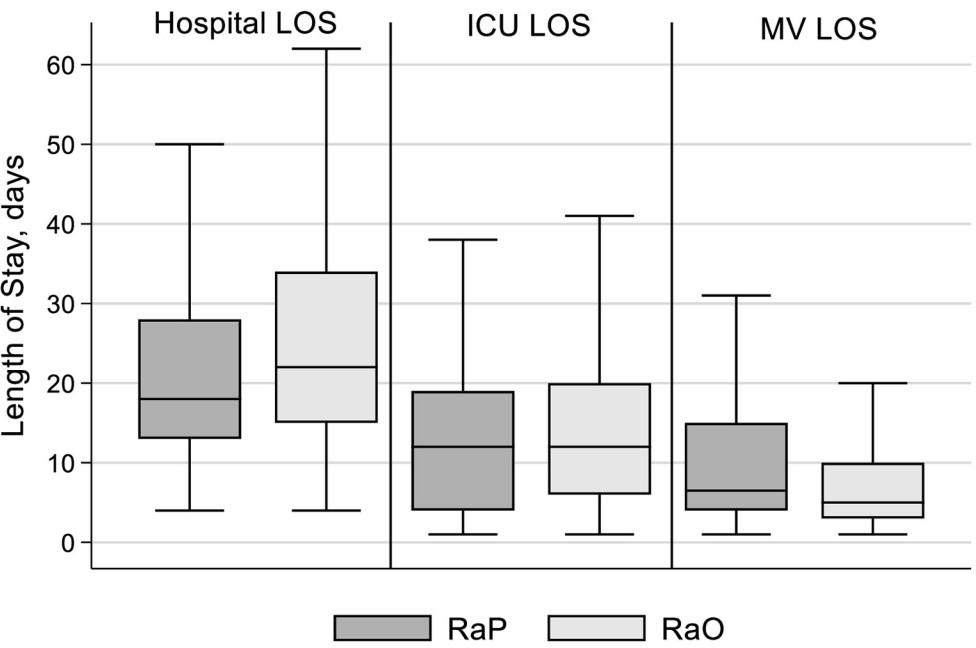

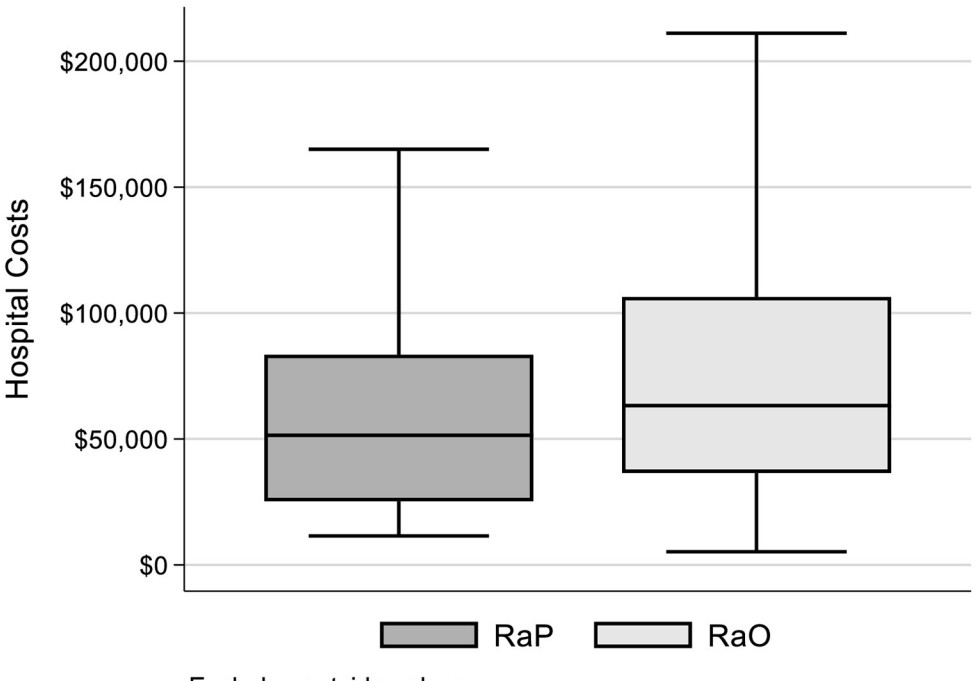

**Fig 3.** Resource use (A) and costs (B) associated with the index hospitalization among patients re-hospitalized due to pneumonia (RaP) or other reasons (RaO). RaP = readmission for pneumonia; RaO = readmission for another condition; LOS = length of stay; ICU = intensive care unit; MV = mechanical ventilation.

streamlined discharge planning and a more coordinated approach to care beyond acute hospitalization would result in better outcomes. While examining these interventions was outside the scope or capacity of our study, a review of previous literature suggests that integrated healthcare delivery across the continuum of care can reduce the need for rehospitalization among certain groups of patients [22]. At the very least, our results may point to a locus of care other than the acute care hospital as the potential place to intervene to save patients additional morbidity.

Our study has a number of limitations and strengths. We were unable to differentiate planned admissions from unplanned ones, thus possibly overestimating the frequency of the patter event. However, these hospital-acquired infections are unlikely to be a major source of planned admissions. As an observational study it is subject to multiple threats to validity, particularly a selection bias. Defining the enrollment criteria prospectively mitigates this bias. Misclassification is of particular concern when using administrative data. To deal with this, we used a previously published, though not clinically validated, algorithm for NP, excluded other potential sources of infection, and included microbiology specimens from specific sources, pharmacy data, and dates of cultures and treatments to minimize its magnitude. If present, however, this type of misclassification would drive the differences between groups toward null. Although confounding is present in all observational studies, we did not attempt to adjust it away, as the aim of the current study was to provide a comparative description of these two groups of patients. As a large multicenter geographically representative database, it is only minimally prone to lack of generalizability. At the same time, our results capture only the events that occur in the hospital, and lack data on post-discharge care. Despite this being the largest and most contemporary multicenter cohort study to examine the epidemiology and outcomes of culture-positive nosocomial pneumonia in the US, a larger sample size would lessen the risk of type II errors in the statistical inferences.

In summary, we have demonstrated that while the risk of rehospitalization among survivors of NP is comparable to that for other types of pneumonia, the rate of repeat admission specifically for pneumonia is low. Indeed, the overwhelming likelihood is that other conditions precipitate the readmission. Furthermore, there is paucity of evidence to support the idea that events surrounding treatment of the index NP have any bearing on the likelihood of having a repeat admission for pneumonia. If confirmed in other studies, this finding may imply that more resources should be allocated to other settings, such as discharge planning, for example, in order to minimize patients' risk for requiring a repeat pneumonia hospitalization.

## Supporting information

**S1 Appendix.**
(XLSX)

## Author Contributions

**Conceptualization:** Marya D. Zilberberg, Laura A. Puzniak, Andrew F. Shorr.

**Data curation:** Brian H. Nathanson.

**Formal analysis:** Brian H. Nathanson.

**Funding acquisition:** Marya D. Zilberberg.

**Investigation:** Noah W. D. Zilberberg.

**Methodology:** Marya D. Zilberberg, Andrew F. Shorr.

**Project administration:** Marya D. Zilberberg.

**Resources:** Laura A. Puzniak.

**Supervision:** Marya D. Zilberberg.

**Validation:** Noah W. D. Zilberberg.

**Writing – original draft:** Marya D. Zilberberg.

**Writing – review & editing:** Marya D. Zilberberg, Brian H. Nathanson, Laura A. Puzniak, Noah W. D. Zilberberg, Andrew F. Shorr.

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
