## [Decision Letter · Decision Letter 0]

12 Jul 2022

PONE-D-22-01197

Descriptive epidemiology of hospitalized patients with bacterial nosocomial pneumonia who experience 30-day readmission in the US, 2014-2019

PLOS ONE

Dear Dr. Zilberberg,

Thank you for submitting your manuscript to PLOS ONE. After careful consideration, we feel that it has merit but does not fully meet PLOS ONE’s publication criteria as it currently stands. Therefore, we invite you to submit a revised version of the manuscript that addresses the points raised during the review process.

The manuscript has been evaluated by a reviewer, and their comments are available below.

The reviewer raised a number of concerns that need attention. They request revisions to the methodological and statistical aspects of the study, in particular with regards to the statistical tests used. They also request more discussion of the potential limitations of the study.

Could you please revise the manuscript to carefully address the concerns raised?

Please note that we have only been able to secure a single reviewer to assess your manuscript. We are issuing a decision on your manuscript at this point to prevent further delays in the evaluation of your manuscript. Please be aware that the editor who handles your revised manuscript might find it necessary to invite additional reviewers to assess this work once the revised manuscript is submitted.

We look forward to receiving your revised manuscript.

Kind regards,

Jamie Royle

Staff Editor

PLOS ONE

https://journals.plos.org/plosone/s/file?id=ba62/PLOSOne_formatting_sample_title_authors_affiliations.pdf".

“Study supported by a grant from Merck Sharp & Dohme Corp., a subsidiary of Merck & Co., Inc., Kenilworth, NJ, USA.

•            MDZ’s employer, EviMed Research Group, LLC, has received research grant support from Merck Sharp & Dohme Corp., a subsidiary of Merck & Co., Inc., Kenilworth, NJ, USA.

•            BHN’s employer, OptiStatim, LLC, has received support from EviMed Research Group, LLC, Goshen, MA.

•            At the time of the study conduct, LAP was an employee of Merck Sharp & Dohme Corp., a subsidiary of Merck & Co., Inc. Kenilworth, NJ, USA and a stockholder in Merck & Co., Inc., Kenilworth, NJ, USA.

•            NWDZ has no conflicts to report.

•            AFS is a consultant to and has received research grant support from Merck Sharp & Dohme Corp., a subsidiary of Merck & Co., Inc., Kenilworth, NJ, USA.”

“This study was supported by a grant from Merck Sharp & Dohme Corp., a subsidiary of Merck & Co., Inc., Kenilworth, NJ, USA.

Potential conflicts of interest

•            MDZ’s employer, EviMed Research Group, LLC, has received research grant support from Merck Sharp & Dohme Corp., a subsidiary of Merck & Co., Inc., Kenilworth, NJ, USA.

•            BHN’s employer, OptiStatim, LLC, has received support from EviMed Research Group, LLC, Goshen, MA.

•            At the time of the study conduct, LAP was an employee of Merck Sharp & Dohme Corp., a subsidiary of Merck & Co., Inc. Kenilworth, NJ, USA and a stockholder in Merck & Co., Inc., Kenilworth, NJ, USA.

•            NWDZ has no conflicts to report.

•            AFS is a consultant to and has received research grant support from Merck Sharp & Dohme Corp., a subsidiary of Merck & Co., Inc., Kenilworth, NJ, USA.”

“This study was supported by a grant from Merck Sharp & Dohme Corp., a subsidiary of Merck & Co., Inc., Kenilworth, NJ, USA.

Potential conflicts of interest

•            MDZ’s employer, EviMed Research Group, LLC, has received research grant support from Merck Sharp & Dohme Corp., a subsidiary of Merck & Co., Inc., Kenilworth, NJ, USA.

•            BHN’s employer, OptiStatim, LLC, has received support from EviMed Research Group, LLC, Goshen, MA.

•            At the time of the study conduct, LAP was an employee of Merck Sharp & Dohme Corp., a subsidiary of Merck & Co., Inc. Kenilworth, NJ, USA and a stockholder in Merck & Co., Inc., Kenilworth, NJ, USA.

•            NWDZ has no conflicts to report.

•            AFS is a consultant to and has received research grant support from Merck Sharp & Dohme Corp., a subsidiary of Merck & Co., Inc., Kenilworth, NJ, USA.”

“I have read the journal's policy.

In addition to funding statement above, the following potential competing interest exist:

•MDZ and AFS have received grant support and/or have served as consultants to Lungpacer, Melinta, Tetraphase, Pfizer, Astellas, Shionogi, The Medicines Company, Spero, and Theravance.”

6. We noted in your submission details that a portion of your manuscript may have been presented or published elsewhere. [Portions of these data have been presented at IDWeek 2021 and ECCMID 2021 annual conference. Other subanalyses of the same cohort have been published or accepted for publication in Critical Care Medicine (citation #11) and Infection Control and Hospital Epidemiology (citation #12).] Please clarify whether this [conference proceeding or publication] was peer-reviewed and formally published. If this work was previously peer-reviewed and published, in the cover letter please provide the reason that this work does not constitute dual publication and should be included in the current manuscript.”

7. In your Data Availability statement, you have not specified where the minimal data set underlying the results described in your manuscript can be found. PLOS defines a study's minimal data set as the underlying data used to reach the conclusions drawn in the manuscript and any additional data required to replicate the reported study findings in their entirety. All PLOS journals require that the minimal data set be made fully available. For more information about our data policy, please see http://journals.plos.org/plosone/s/data-availability.

8. We note that you have indicated that data from this study are available upon request. PLOS only allows data to be available upon request if there are legal or ethical restrictions on sharing data publicly. For more information on unacceptable data access restrictions, please see http://journals.plos.org/plosone/s/data-availability#loc-unacceptable-data-access-restrictions.

Reviewers' comments:

Reviewer's Responses to Questions

**Comments to the Author**

1. Is the manuscript technically sound, and do the data support the conclusions?

Reviewer #1: Yes

2. Has the statistical analysis been performed appropriately and rigorously? 

Reviewer #1: No

3. Have the authors made all data underlying the findings in their manuscript fully available?

Reviewer #1: No

4. Is the manuscript presented in an intelligible fashion and written in standard English?

Reviewer #1: Yes

5. Review Comments to the Author

Reviewer #1: This study uses the Premier database to identify a cohort of patients with pneumonia and examine 30-day readmission rates, with stratification for patients who were readmitted specifically for pneumonia versus other reasons, and identify differences in these readmitted groups. Overall, the authors found that 15.2% of patients admitted for pneumonia required readmission within 30 days. Of these, 4.9% were readmitted specifically for pneumonia while the remainder were readmitted for other reasons. Several differences were found between these two subgroups that might inform decisions about how to prevent readmission in patients admitted for pneumonia. Overall, the study design is sound, the data source is reliable and well respected, and the results are intuitive even if minimally informative, and the paper is well written.

Major Issues

Many patients have planned readmissions within 30 days of discharge. The authors should comment on planned versus unplanned readmission. If the data do not allow such a distinction, this should be noted as a limitation.

Many patients who are readmitted do not return to the original hospital. How was this handled? If the authors cannot tell when a patient was readmitted to the originating hospital, this should be noted in the definition of readmission, and also noted as a limitation.

Page 4, Line 6. The authors state that the study was exempt from ethics review. Who made this determination? An IRB needs to make this determination. If an IRB deemed this study exempt, the authors need to state the IRB that made this determination. If the authors themselves declared this study exempt, then I do not believe the study has been properly vetted for human subjects protection.

Methods. Statistical comparisons were made using ANOVA and Kruskall-Wallace tests. However, these are two-group comparisons not 3+ group comparisons. I think t tests and chi-square tests should be used for parametric comparisons of continuous and binary/categorical variables, respectively, and Mann-Whitney and chi-square tests should be used for non-parametric tests of continuous and binary/categorical variables, respectively.

Methods. Statistical tests have been applied to each category for categorical variables for some variable in Table 2. Culture source and IET contains categories that are exhaustive and mutually exclusive. I think there should be a single omnibus test for these variables, not individual tests applied to each category within the variable.

Minor Issues

Page 3, Line 2. Most sources use the HAC abbreviation for hospital acquired conditions rather than hospital acquired complication. Consider using the standard abbreviation used in the infectious disease literature.

Page 11, first paragraph states that “While the ICU LOS, did not differ between the two groups…”. This is incorrect. The difference in ICU LOS was not significant, but that is not the same thing as saying that they did not differ. To say they did not differ implies that the mean and all values are exactly identical, which is not true. Consider saying that mean ICU LOS did not differ significantly.

Page 13, second paragraph. The use of language around RaP and RaO is confusing. For example, the authors state that “there was a significantly higher prevalence of nvHABP among those with RaP than those with RaO”. This sounds like RaP and RaO are diseases or conditions. Later in the paragraph the authors say “the MV duration was longer in the setting of RaP than in the group with RaO”. RaP and RaO are not settings. Consider using language that clearly indicates you are talking about patients with specific types of readmissions.

Page 15, first paragraph. The authors argue that lack of generalizability is of minimal concern. I disagree. The only setting to which the results can be reasonably generalized is populations that reflect the Premier database. Thus, it cannot be generalized to an individual hospital, a single state, or the US more broadly. I think generalizability is a limitation and should be acknowledged and not minimized.

6. PLOS authors have the option to publish the peer review history of their article (what does this mean?). If published, this will include your full peer review and any attached files.

Reviewer #1: No

---

## [Author Response · Author response to Decision Letter 0]

5 Aug 2022

July 20, 2022

Dear Dr. Chenette,

We are submitting a revised version of our manuscript “Descriptive epidemiology of hospitalized patients with bacterial nosocomial pneumonia who experience 30-day readmission in the US, 2014-2019” along with our point-by-point responses to the reviewer’s comments below. We hope we have responded adequately to all of the concerns, and have corrected all the points to comply with the editorial style.

We are grateful to the reviewer for their careful attention to our work, and believe addressing their concerns has strengthened our manuscript. 

We are looking forward to your further thoughts. 

Respectfully,

Marya Zilberberg, MD, MPH, on behalf of my co-authors 

Responses

Data availability

Premier is a proprietary database available for purchase to any investigator. Because of the legal constraints of inlicensing, we are unable to upload the data we used. 

Funding

This study was supported by a grant from Merck Sharp & Dohme Corp., a subsidiary of Merck & Co., Inc., Kenilworth, NJ, USA.

• MDZ’s employer, EviMed Research Group, LLC, has received research grant support from Merck Sharp & Dohme Corp., a subsidiary of Merck & Co., Inc., Kenilworth, NJ, USA.

• BHN’s employer, OptiStatim, LLC, has received support from EviMed Research Group, LLC, Goshen, MA.

• At the time of the study conduct, LAP was an employee of Merck Sharp & Dohme Corp., a subsidiary of Merck & Co., Inc. Kenilworth, NJ, USA and a stockholder in Merck & Co., Inc., Kenilworth, NJ, USA.

• NWDZ has no conflicts to report.

• AFS is a consultant to and has received research grant support from Merck Sharp & Dohme Corp., a subsidiary of Merck & Co., Inc., Kenilworth, NJ, USA.

https://journals.plos.org/plosone/s/file?id=ba62/PLOSOne_formatting_sample_title_authors_affiliations.pdf".

AU: Thank you. 

AU: This is because the submission website menu does not include the option for our specific funding source as articulated in the Disclosure sections of our manuscript. We would be grateful if the Editorial office could correct this on the website to match our paper. 

“Study supported by a grant from Merck Sharp & Dohme Corp., a subsidiary of Merck & Co., Inc., Kenilworth, NJ, USA.

• MDZ’s employer, EviMed Research Group, LLC, has received research grant support from Merck Sharp & Dohme Corp., a subsidiary of Merck & Co., Inc., Kenilworth, NJ, USA.

• BHN’s employer, OptiStatim, LLC, has received support from EviMed Research Group, LLC, Goshen, MA.

• At the time of the study conduct, LAP was an employee of Merck Sharp & Dohme Corp., a subsidiary of Merck & Co., Inc. Kenilworth, NJ, USA and a stockholder in Merck & Co., Inc., Kenilworth, NJ, USA.

• NWDZ has no conflicts to report.

• AFS is a consultant to and has received research grant support from Merck Sharp & Dohme Corp., a subsidiary of Merck & Co., Inc., Kenilworth, NJ, USA.”

AU: Although LAP participated in the design and reporting of the study, the company had no role in study design, data collection and analysis, decision to publish, or preparation of the manuscript. 

“This study was supported by a grant from Merck Sharp & Dohme Corp., a subsidiary of Merck & Co., Inc., Kenilworth, NJ, USA.

Potential conflicts of interest

• MDZ’s employer, EviMed Research Group, LLC, has received research grant support from Merck Sharp & Dohme Corp., a subsidiary of Merck & Co., Inc., Kenilworth, NJ, USA.

• BHN’s employer, OptiStatim, LLC, has received support from EviMed Research Group, LLC, Goshen, MA.

• At the time of the study conduct, LAP was an employee of Merck Sharp & Dohme Corp., a subsidiary of Merck & Co., Inc. Kenilworth, NJ, USA and a stockholder in Merck & Co., Inc., Kenilworth, NJ, USA.

• NWDZ has no conflicts to report.

• AFS is a consultant to and has received research grant support from Merck Sharp & Dohme Corp., a subsidiary of Merck & Co., Inc., Kenilworth, NJ, USA.”

“This study was supported by a grant from Merck Sharp & Dohme Corp., a subsidiary of Merck & Co., Inc., Kenilworth, NJ, USA.

Potential conflicts of interest

• MDZ’s employer, EviMed Research Group, LLC, has received research grant support from Merck Sharp & Dohme Corp., a subsidiary of Merck & Co., Inc., Kenilworth, NJ, USA.

• BHN’s employer, OptiStatim, LLC, has received support from EviMed Research Group, LLC, Goshen, MA.

• At the time of the study conduct, LAP was an employee of Merck Sharp & Dohme Corp., a subsidiary of Merck & Co., Inc. Kenilworth, NJ, USA and a stockholder in Merck & Co., Inc., Kenilworth, NJ, USA.

• NWDZ has no conflicts to report.

• AFS is a consultant to and has received research grant support from Merck Sharp & Dohme Corp., a subsidiary of Merck & Co., Inc., Kenilworth, NJ, USA.”

AU: We have removed all funding related language from the manuscript. 

“I have read the journal's policy.

In addition to funding statement above, the following potential competing interest exist:

•MDZ and AFS have received grant support and/or have served as consultants to Lungpacer, Melinta, Tetraphase, Pfizer, Astellas, Shionogi, The Medicines Company, Spero, and Theravance.”

AU: Done. 

6. We noted in your submission details that a portion of your manuscript may have been presented or published elsewhere. [Portions of these data have been presented at IDWeek 2021 and ECCMID 2021 annual conference. Other subanalyses of the same cohort have been published or accepted for publication in Critical Care Medicine (citation #11) and Infection Control and Hospital Epidemiology (citation #12).] Please clarify whether this [conference proceeding or publication] was peer-reviewed and formally published. If this work was previously peer-reviewed and published, in the cover letter please provide the reason that this work does not constitute dual publication and should be included in the current manuscript.”

AU: We have added the following to this section: “Analyses presented in the current manuscript have not undergone previous peer review or publication.”

7. In your Data Availability statement, you have not specified where the minimal data set underlying the results described in your manuscript can be found. PLOS defines a study's minimal data set as the underlying data used to reach the conclusions drawn in the manuscript and any additional data required to replicate the reported study findings in their entirety. All PLOS journals require that the minimal data set be made fully available. For more information about our data policy, please see http://journals.plos.org/plosone/s/data-availability.

AU: As we stated in our manuscript, “The data used in this study derive from Premier Research database, a proprietary third-party database available to researchers through a specific agreement with Premier.” This does not seem to be one of your “unacceptable data restrictions” as found on your website here: https://journals.plos.org/plosone/s/data-availability#loc-unacceptable-data-access-restrictions

In fact, anyone can obtain the data in the same manner that we did. However, since this is a proprietary database, albeit available for purchase to any investigators, because of the legalities of inlicensing, we are unable to upload the data we used. 

8. We note that you have indicated that data from this study are available upon request. PLOS only allows data to be available upon request if there are legal or ethical restrictions on sharing data publicly. For more information on unacceptable data access restrictions, please see http://journals.plos.org/plosone/s/data-availability#loc-unacceptable-data-access-restrictions.

AU: See above

Reviewers' comments:

Reviewer's Responses to Questions

Comments to the Author

1. Is the manuscript technically sound, and do the data support the conclusions?

Reviewer #1: Yes

2. Has the statistical analysis been performed appropriately and rigorously?

Reviewer #1: No

3. Have the authors made all data underlying the findings in their manuscript fully available?

Reviewer #1: No

4. Is the manuscript presented in an intelligible fashion and written in standard English?

Reviewer #1: Yes

5. Review Comments to the Author

Reviewer #1: This study uses the Premier database to identify a cohort of patients with pneumonia and examine 30-day readmission rates, with stratification for patients who were readmitted specifically for pneumonia versus other reasons, and identify differences in these readmitted groups. Overall, the authors found that 15.2% of patients admitted for pneumonia required readmission within 30 days. Of these, 4.9% were readmitted specifically for pneumonia while the remainder were readmitted for other reasons. Several differences were found between these two subgroups that might inform decisions about how to prevent readmission in patients admitted for pneumonia. Overall, the study design is sound, the data source is reliable and well respected, and the results are intuitive even if minimally informative, and the paper is well written.

Major Issues

Many patients have planned readmissions within 30 days of discharge. The authors should comment on planned versus unplanned readmission. If the data do not allow such a distinction, this should be noted as a limitation.

AU: We appreciate the reviewer’s attention to this point. However, we are not aware of any evidence that “many patients have planned readmissions” among survivors of a hospitalization involving HAP or VAP. We would be grateful to the reviewer if they could share citations on the incidence of this event in this population of patients, so that we know how to assess its potential impact on our findings. 

Many patients who are readmitted do not return to the original hospital. How was this handled? If the authors cannot tell when a patient was readmitted to the originating hospital, this should be noted in the definition of readmission, and also noted as a limitation.

AU: The reviewer is quite correct on this point, and we are grateful for the opportunity to add this to the manuscript.

Methods section, page 4: “All readmissions could be identified only if the patient presented to the same hospital where their index infection had occurred.”

Limitations paragraph in the Discussion section, page 15: “Another potential source of misclassification pertains to the definition of readmission. As indicated in the Methods, the structure of the database does not allow to identify readmissions to a hospital distinct from the one where the index hospitalization occurred. For this reason, we may have underestimated the magnitude of 30-day hospitalizations. However, this misclassification is also likely non-differential, thus potentially reducing the observed differences between the two groups we examined.” 

Page 4, Line 6. The authors state that the study was exempt from ethics review. Who made this determination? An IRB needs to make this determination. If an IRB deemed this study exempt, the authors need to state the IRB that made this determination. If the authors themselves declared this study exempt, then I do not believe the study has been properly vetted for human subjects protection.

AU: It has been our understanding that deidentified information provided for secondary research in Premier database meets the criteria for exemption. We invite the reviewer to examine the HHS website page from which our understanding derives (https://www.hhs.gov/ohrp/regulations-and-policy/decision-charts-2018/index.html#c1). We should also mention that in our past efforts this database has been deemed exempt. 

Methods. Statistical comparisons were made using ANOVA and Kruskall-Wallace tests. However, these are two-group comparisons not 3+ group comparisons. I think t tests and chi-square tests should be used for parametric comparisons of continuous and binary/categorical variables, respectively, and Mann-Whitney and chi-square tests should be used for non-parametric tests of continuous and binary/categorical variables, respectively.

AU: We thank the reviewer for this correction. The reviewer is correct in that t-tests, Mann-Whitney tests, and Chi-Square tests are appropriate, and indeed were the tests used in the analysis. We apologize for incorrectly describing the Methods based on a prior analysis of the data where we compared the 3 pneumonia groups. We have now corrected the Methods description to reflect what was done and to align with the reviewer’s comments. 

Methods. Statistical tests have been applied to each category for categorical variables for some variable in Table 2. Culture source and IET contains categories that are exhaustive and mutually exclusive. I think there should be a single omnibus test for these variables, not individual tests applied to each category within the variable.

AU: The reviewer brings up an interesting point. In Table 2, we do provide a single p-value for IET status. Culture source in our data was a little tricky. We had a small number of patients who had multiple positive cultures taken on the same “index day” (e.g., a positive sputum and respiratory culture). Consequently, we are presenting them as individual variables and we note that the differences between groups were modest regardless of the p-values. This comment did, however, incited us to revise the way we are presenting the results of polymicrobial infections. We are now using one omnibus p-value for this variable. Thank you for this suggestion. 

Minor Issues

Page 3, Line 2. Most sources use the HAC abbreviation for hospital acquired conditions rather than hospital acquired complication. Consider using the standard abbreviation used in the infectious disease literature.

AU: Changed to “condition.”

Page 11, first paragraph states that “While the ICU LOS, did not differ between the two groups…”. This is incorrect. The difference in ICU LOS was not significant, but that is not the same thing as saying that they did not differ. To say they did not differ implies that the mean and all values are exactly identical, which is not true. Consider saying that mean ICU LOS did not differ significantly.

AU: We have added the word “significantly,” thank you. 

Page 13, second paragraph. The use of language around RaP and RaO is confusing. For example, the authors state that “there was a significantly higher prevalence of nvHABP among those with RaP than those with RaO”. This sounds like RaP and RaO are diseases or conditions. Later in the paragraph the authors say “the MV duration was longer in the setting of RaP than in the group with RaO”. RaP and RaO are not settings. Consider using language that clearly indicates you are talking about patients with specific types of readmissions.

AU: We have taken the reviewer’s advice and clarified this point in multiple places in the paragraph in question. 

Page 15, first paragraph. The authors argue that lack of generalizability is of minimal concern. I disagree. The only setting to which the results can be reasonably generalized is populations that reflect the Premier database. Thus, it cannot be generalized to an individual hospital, a single state, or the US more broadly. I think generalizability is a limitation and should be acknowledged and not minimized.

AU: We are not sure we understand the reviewer’s statement. In our understanding of the terminology, “generalizability” or “external validity” pertains to a general theory of the case, rather than specific subpopulations. In this way, Premier is fairly representative of the US landscape of the population we have defined. Therefore, we state that the study is generalizable. We do, however, agree that, as in all studies like this, it is difficult to apply findings to individual hospitals or other subgroupings. 

6. PLOS authors have the option to publish the peer review history of their article (what does this mean?). If published, this will include your full peer review and any attached files.

Do you want your identity to be public for this peer review? For information about this choice, including consent withdrawal, please see our Privacy Policy.

Reviewer #1: No

---

## [Decision Letter · Decision Letter 1]

8 Sep 2022

PONE-D-22-01197R1Descriptive epidemiology of hospitalized patients with bacterial nosocomial pneumonia who experience 30-day readmission in the US, 2014-2019PLOS ONE

Dear Dr. Zilberbeg,

Thank you for submitting your manuscript to PLOS ONE. After careful consideration, we feel that it has merit but does not fully meet PLOS ONE’s publication criteria as it currently stands. Therefore, we invite you to submit a revised version of the manuscript that addresses the points raised during the review process. Please address the two final comments by the reviewer. Mainly, regarding ethical approval:As the requirements of journals and institutions are becoming stricter, approval from an independent ethics committee is becoming the norm for all research studies involving human participants and/or medical information independently of how low the risks are.

Could you please provide further details on why your study is exempt from the need for approval and provide confirmation from your institutional review board or research ethics committee (e.g., in the form of a letter or email correspondence) that ethics review was not necessary for this study? Please include a copy of the correspondence as an "Other" file.

We look forward to receiving your revised manuscript.

Kind regards,

Dafna Yahav

Academic Editor

PLOS ONE

Journal Requirements:

Reviewers' comments:

Reviewer's Responses to Questions

**Comments to the Author**

1. If the authors have adequately addressed your comments raised in a previous round of review and you feel that this manuscript is now acceptable for publication, you may indicate that here to bypass the “Comments to the Author” section, enter your conflict of interest statement in the “Confidential to Editor” section, and submit your "Accept" recommendation.

Reviewer #1: (No Response)

2. Is the manuscript technically sound, and do the data support the conclusions?

Reviewer #1: Yes

3. Has the statistical analysis been performed appropriately and rigorously? 

Reviewer #1: Yes

4. Have the authors made all data underlying the findings in their manuscript fully available?

Reviewer #1: No

5. Is the manuscript presented in an intelligible fashion and written in standard English?

Reviewer #1: Yes

6. Review Comments to the Author

Reviewer #1: The authors have been responsive to much of the prior round of peer review. The following issues remain.

1. The authors have resisted adding some language acknowledging that some readmissions are planned, and therefore their readmission rates may exaggerate the problem. Even CMS’s Hospital Readmission Reduction Program, which includes Pneumonia as a target, attempts to account for planned versus unplanned readmission (https://qualitynet.cms.gov/inpatient/hrrp/measures). This seems ample evidence that it is at least a potential limitation worth mentioning. I will leave it to the editor to decide whether this is a reasonable addition to the limitations section.

2. The authors have made their own determination that this study is exempt from IRB review. The reason we have IRBs and offices of human subjects protection is so that these decisions can be made independently. Obviously, the authors can read a flowchart for human subjects exemption. But, in my opinion they do not have the authority to make their own declaration. They should submit their study design to an independent IRB, who will certify that their research is exempt from full review. I will leave it to the editor to decide whether this is a reasonable request.

Otherwise, I am satisfied with the authors' other responses to issues raised in the first round of peer review.

7. PLOS authors have the option to publish the peer review history of their article (what does this mean?). If published, this will include your full peer review and any attached files.

Reviewer #1: No

---

## [Author Response · Author response to Decision Letter 1]

14 Sep 2022

Reviewer #1: The authors have been responsive to much of the prior round of peer review. The following issues remain.

1. The authors have resisted adding some language acknowledging that some readmissions are planned, and therefore their readmission rates may exaggerate the problem. Even CMS’s Hospital Readmission Reduction Program, which includes Pneumonia as a target, attempts to account for planned versus unplanned readmission (https://qualitynet.cms.gov/inpatient/hrrp/measures). This seems ample evidence that it is at least a potential limitation worth mentioning. I will leave it to the editor to decide whether this is a reasonable addition to the limitations section.

AU: We have now added the following to the Limitations paragraph on page 14:

“We were unable to differentiate planned admissions from unplanned ones, thus possibly overestimating the frequency of the patter event. However, these hospital-acquired infections are unlikely to be a major source of planned admissions.”

2. The authors have made their own determination that this study is exempt from IRB review. The reason we have IRBs and offices of human subjects protection is so that these decisions can be made independently. Obviously, the authors can read a flowchart for human subjects exemption. But, in my opinion they do not have the authority to make their own declaration. They should submit their study design to an independent IRB, who will certify that their research is exempt from full review. I will leave it to the editor to decide whether this is a reasonable request.

AU: We appreciate the reviewer’s strong attention to the ethics of study contact consideration. Below we present the rationale for why our study does not require such. We also attach as Supporting Information a letter from a qualified statistician who reviewed Premier methodologies for de-identifying patient data. 

Since our study is secondary research using de-identified database, it is not considered as human subjects research. Our group has published dozens of manuscripts using the Premier healthcare database in various peer-reviewed journals. IRB review was never required. As we refence in the citation #9, our study falls squarely into “research not involving human subjects” under 45 CFR Part 46, which can be found here: https://www.hhs.gov/ohrp/regulations-and-policy/decision-charts-2018/index.html#c1

---

## [Decision Letter · Decision Letter 2]

2 Oct 2022

Descriptive epidemiology of hospitalized patients with bacterial nosocomial pneumonia who experience 30-day readmission in the US, 2014-2019

PONE-D-22-01197R2

Dear Dr. Zilberberg,

We’re pleased to inform you that your manuscript has been judged scientifically suitable for publication and will be formally accepted for publication once it meets all outstanding technical requirements.

Kind regards,

Steven Eric Wolf, MD

Academic Editor

PLOS ONE

Additional Editor Comments (optional):

Reviewers' comments:

Reviewer's Responses to Questions

**Comments to the Author**

1. If the authors have adequately addressed your comments raised in a previous round of review and you feel that this manuscript is now acceptable for publication, you may indicate that here to bypass the “Comments to the Author” section, enter your conflict of interest statement in the “Confidential to Editor” section, and submit your "Accept" recommendation.

Reviewer #1: All comments have been addressed

2. Is the manuscript technically sound, and do the data support the conclusions?

Reviewer #1: Yes

3. Has the statistical analysis been performed appropriately and rigorously? 

Reviewer #1: Yes

4. Have the authors made all data underlying the findings in their manuscript fully available?

Reviewer #1: Yes

5. Is the manuscript presented in an intelligible fashion and written in standard English?

Reviewer #1: Yes

6. Review Comments to the Author

Reviewer #1: I am satisfied with the responses to the last round of peer review.

7. PLOS authors have the option to publish the peer review history of their article (what does this mean?). If published, this will include your full peer review and any attached files.

Reviewer #1: No

---

## [Editor Report · Acceptance letter]

9 Nov 2022

PONE-D-22-01197R2 

Descriptive epidemiology of hospitalized patients with bacterial nosocomial pneumonia who experience 30-day readmission in the US, 2014-2019 

Dear Dr. Zilberberg:

I'm pleased to inform you that your manuscript has been deemed suitable for publication in PLOS ONE. Congratulations! Your manuscript is now with our production department. 

Kind regards, 

on behalf of

Dr. Steven Eric Wolf 

Academic Editor

PLOS ONE